# Learning-oriented motivation: Examining the impact of teaching practices with motivational potential

Jorge Valenzuela[1,2]*, Jorge Miranda-Ossandon[3], Carla Muñoz[2,4], Andrea Precht[1], Milenko Del Valle[5], Juan-Gabriel Vergaño-Salazar[6]

1 Facultad de Ciencias de la Educación, Universidad Católica del Maule, Talca, Chile, 2 Millennium Nucleus for the Science of Learning, Talca, Chile, 3 Facultad de Educación, Universidad Católica de Temuco, Temuco, Chile, 4 Departamento de Psicología, Universidad Católica del Maule, Talca, Chile, 5 Facultad de Ciencias Sociales, Universidad de Antofagasta, Antofagasta, Chile, 6 Departamento de Análisis de Datos, Universidad Autónoma de Chile, Talca, Chile

☯ These authors contributed equally to this work.
* jvalenzuela@ucm.cl

**Data Availability Statement:** The data underlying the results of this article are available at https://doi.org/10.5281/zenodo.10427851.

**Funding:** This research was funded by Agencia Nacional de Investigación y Desarrollo, ANID –

## Abstract

This study compares the predictive ability of nine different types of motivational practices on the motivational orientation toward learning. Given the nature of undergraduate studies, identifying the most predictive motivational variables on learning orientation allows us to focus our efforts on those motivational practices to guide students to deploy their cognitive resources by focusing on learning and not only on obtaining good grades. The study included Chilean university students from health (n = 398) and education (n = 365) programs. A Bayesian multiple regression was carried out in both groups. The results show strong evidence of a specific effect of motivational practices on motivational orientation towards learning. Although the impact on motivational orientation toward learning may vary slightly across different fields of study, the primary predictors consistently are practices that emphasize importance and foster autonomy. The effect of utility-focused motivational practices is observed only within the predictive model for the group of health students.

## Introduction

There is consensus that motivation plays a relevant role in learning [1, 2] and that different factors influence it [e.g., self-determination, self-efficacy, self-regulation]. However, it should be cautioned that identifying these factors depends on the theoretical approaches adopted [3]. Thus, in a general way, "motivation" is characterized as the intensity of that force that initiates, maintains, orients, or allows the choice of a specific behavior. However, more than the intensity of that force is required. Suppose we want to assess its effect as a learning factor in depth. In that case, It is also necessary to consider the type of motivational orientation, that is, the process of directing or guiding a person's motivations toward a specific goal (learning or performance) [4].

Chile, through the Regular Fondecyt project 1210626 [JV] and the support of Millennium Nucleus for the Science of Learning NCS2022_26 [CM, JV].

**Competing interests:** The authors have declared that no competing interests exist.

Motivation toward learning can be conceptualized from different theories, and in each of them, we found variables that could shed light on the learning process. Perhaps the most robust theories that have made a greater contribution to the understanding of motivation towards learning are the Self-Determination Theory [Cf. 5–7]; the Socio-Cognitive Theory, whose central element is self-efficacy [8–10]; The Expectancy and Value Theory [Cf. ETV in 11–13]; and, finally, the Goal Theory [Cf. GT in 4, 14–16].

Although all these theories have a significant predictive capacity concerning general outcomes, Goal Theory (GT) has the additional strength of focusing on a specific type of motivated action: willingness to learn. This is our very focus of interest: to understand how students activate their cognitive, volitional, metacognitive, and motivational resources to orient them towards learning, that is, towards the appropriation and integration of the contents and skills proposed in the university educational context.

In this perspective, Goal Theory [3, 17–19] identifies at least three types of motivational orientations related to learning: (a) motivational performance orientation in its approach mode (wanting to have a good result), (b) in its avoidance mode to avoid failure [18], and finally, (c) a motivational learning orientation, understood as content mastery.

In this context, we are particularly interested in the motivational orientation toward learning or mastery goals [4, 20]. This option arises from the realization that it is not enough to be "motivated" in higher education since much of this motivation is only oriented towards performance and translates into fulfilling what is necessary to advance in the certification process imposed by higher education. For its part, learning-oriented motivation or mastery goals are associated with deep learning strategies [21, 22], self-regulation of learning [23] and augur a better quality in the learning of skills, knowledge and professional competencies that are supposed to be acquired in university education.

However, although this approach to the motivational phenomenon allows a foresight view of the consequences of orienting motivation in one direction or another [2], more evidence is needed about the motivational teachers' practices that would determine whether a student develops a learning-oriented motivation. These motivational practices can be identified from multiple theoretical perspectives, and the current literature shows the benefits of integrating different theories to explain this phenomenon [24, 25].

## Types of motivational practices deployed by university teachers

The practices that teachers use to promote learning can be qualified as motivational to the extent that there is, at least, a theoretical basis for assuming that they have such an effect and, ideally, empirical evidence of this effect. In this sense, motivational theories help us identify teaching practices with motivational potential.

Self-determination theory recognizes that motivation is influenced by satisfying three basic psychological needs: autonomy, competence, and relatedness [26]. Likewise, motivation can be regulated internally or externally. In the latter case, through **rewards** or **punishments** [5]. Similarly, we know that intrinsic motivation arises from a process of internalization of motives and self-determination, predominantly mediated by experiences of **autonomy** [27] and that although both types of motivation have effects on learning, they contribute in different ways. So, both a dynamic of rewards and punishments and the promotion of self-determination (autonomy) would be important elements to explore to understand why students develop an internal or external motivational orientation toward learning [Cf. 28].

Social-cognitive Theory, on the other hand, shows us that the feeling of **self-efficacy** is a determinant for the choice of goals, and we have evidence that it plays a crucial role in the development of goal orientation [see mastery goal in 29]. This factor, viewed from a teaching

practice perspective, can translate into experiences of cognitive **challenge**, where the teacher implicitly expresses his or her beliefs that the student can solve the task successfully. The Expectancy/Value Theory (EVT) also includes a variable equivalent to self-efficacy in its expectancy component and adds the task value as an explanatory variable. This last dimension comprises perceived task **utility**, **importance**, **cost**, and **interest**. Importance and utility correspond to subdimensions of the value of the task [12, 30, 31] and differ in that while importance expresses the value in terms of more identity-related aspects of the person, the utility does so in terms of more instrumental, external and mediated aspects, such as futures plans [32]. Cost, on the other hand, expresses the value of the task in terms of how much one is willing to sacrifice. Finally, the interest expresses the task value through the pleasure that the activity generates.

In addition to the above factors, we can identify two others that act as a necessary but not sufficient condition for learning motivation. On the one hand, those teaching practices of emotional support and containment for the student, individually, which we call **emotional support** practices, and those that are carried out collectively and which are expressed as the creation of an adequate and **safe classroom climate** for learning. Recent reports linked to online teaching show that students identify these emotional support factors as crucial in their desire to learn [33–37].

In this context, the various theories of motivation allow complementary approaches [e.g., 38], in our case, to understand learning orientation. Goal theory allows us to distinguish this orientation from others focused on performance and whether this orientation is built from avoidance or from approaching that goal. In turn, Self-determination theory permits us to categorize by the type of regulation (internal or external) the motivational practices that teachers carry out to motivate their students. The motives that drive academic practices can help us identify those that significantly contribute to the desire to learn, not just academic performance. The Expectancy-value theory enables us to examine teaching practices that ultimately affect learning-oriented motivation. It does so by focusing on the specific qualifications of the motivational object, such as mastery, and distinguishing between utility (instrumental value) and importance (value linked to identity-related aspects) as specific dimensions of the value attributed to the learning task.

The aim here is not to make the different theories compete in terms of their explanatory capacity or precision but to take advantage of the nuances and perspectives of these theories in terms of utility [24] to identify those practices that best support decision-making at the educational level, in this case, concerning favoring practices that support a motivational orientation towards learning or mastery.

## Problem

Numerous authors highlight the distinctive features of careers in health and education, emphasizing their strong service orientation and societal significance [39, 40]. Nevertheless, these fields exhibit formative differences [41–43]. Within this context, the identification and characterization of motivational teaching practices become crucial, enabling pinpointing those practices that exert the most significant influence, not on motivation in general but specifically on the orientation of motivation toward learning.

Indeed, we have evidence of motivational factors linked to learning [1, 44], but we need to know which of the motivational practices developed by university teachers favor learning-oriented motivation. The present study identifies and analyzes the effect of motivational teaching practices explicitly linked to rewards, punishments, utility, importance, self-efficacy, emotional support, classroom climate, and autonomy development on motivational orientation toward learning.

## Method

### Participants

Chile has a mixed system of private and public universities regulated by a state accreditation agency, and they are responsible for awarding professional and bachelor's degrees. Universities are mainly accessed through a unique admission system that considers a standardized national selection test (PAES) and the ranking of school grades. Within the university system in 2022 (54.8% female), health careers account for 19.6% of the total enrolment (60,243), while education careers account for 9.4% of university students (27,832) [45].

Students from public and private Chilean university programs in Health (n = 398) and Education (n = 365), female = 599 (79%), male = 151 (19.9%), other = 8 (1.1%), participated in the present study. Although the percentage of women is high (79%), this is a situation that is common in highly feminized programs [Cf. 46]. The average age of the participants was 20.9 years (SD = 3.39), concentrated in the 18–24 years age group (91.1% of the sample).

The health area included Medicine, Nursing, Kinesiology, Occupational Therapy, Psychology, Chemistry and Pharmacy, Medical Bioengineering, Nutrition, Dentistry, Obstetrics, and Speech Therapy. The education area considered pedagogical programs in preschool, elementary, and pedagogy in different specialties: Mathematics, Language, Religion and Philosophy, English, History, Science, and Physical Education.

### Instruments

The Motivational Potential of Practices (MPP) was evaluated for nine types of motivational practices: rewards, punishment, autonomy, challenge, utility, task importance, interest, emotional support, and safety ambience. The authors developed this instrument based on previous research (47] and applied it in its original Spanish version (see S1 Appendix). The practices evaluated are related to the aspects that the literature recognizes as factors influencing motivation. Each of these dimensions was measured through items that account for prototypical practices of each of these dimensions [Cf. 47], through Likert scales (0–5) in two dimensions: frequency of the practice (FP) and Motivational Effect attributed to that Practice (MPE). The frequency was measured through the question "How often do your teachers perform the following practices. . ." (0 = never—5 = always). The Motivational Effect attributed to this practice, through the item: The following practices of the teachers generate in me a desire to learn in the program (0 = Strongly disagree 5 = Strongly agree). From these two variables, the variable Motivational Potential of the Practices (MPP) was constructed and corresponded to the square root of the product between frequency and the motivational effect attributed to each type of practice ($MPP = \sqrt{PF*MPE}$).

Although the different motivational potential practices are grouped into a single factor, confirming their motivational role ($Chi2_{(13)} = 52.7$, p< .001; CFI = .981; TLI = .970; SRMR = .031; RMSEA = .06 [.04, .08]), and with good reliability index (a = .809; ω = .866), we were interested in identifying in a specific way which of these motivational practices have a greater impact on the motivational orientation towards learning. Thus, although each type of motivational practice can affect the motivational orientation towards learning, it is essential to identify which of them has a greater explanatory capacity in each of the groups of students studied (health and education).

Motivational orientation towards learning was measured using the Mastery Goal subscale of the Achievement Goal Scale [48] through a 6-point Likert scale. In the present study, this scale showed good psychometric characteristics concerning validity ($Chi^2_{(335)} = 112$; p = .001; CFI = .939; TLI = .926; SMRS = .063; RMSEA = .053 [.04, .06]) and reliability (Cronbach' α =. 800; McDonald' ω = .817).

It is essential to clarify that while there may be a relationship between students' statements about which practices motivate them most and their actual motivational orientation towards learning, the two measures point to different motivational dimensions. The former is situated in a general motivational framework, and the second focuses on a specific type of motivation: learning-oriented motivation.

## Procedures

Data collection was performed online through an online platform via institutional invitations to participate in the study. Before accessing the instrument, the participant had to declare his/her agreement with the informed consent approved by the Scientific Ethics Committee of the sponsoring institution. Access to the participants was through e-mail and institutional invitations in different faculties of Education and Health throughout the country. For legal reasons, only those over 18 years of age were allowed to participate. Data collection was conducted online between 4 April and 29 May 2022. Informed consent was accepted electronically. Access to the survey was granted only after acceptance of participation.

## Analytical procedures

The prior verification of the psychometric characteristics of the scales was carried out through confirmatory factor analysis (CFA) and reliability analysis. For the CFA fit indices were used: Chi2, Comparative Fit Index (CFI), Tucker-Lewis Index (TLI), Standardized Root Mean Square Residual (SRMR), and Root Mean Square Error of Approximation (RMSEA) and their respective confidence intervals. The parameters for each of them were evaluated through the criteria provided by the parameters proposed by the literature (e.g., chi2 > .05, RMSEA < .08; TLI > .90 CFI > .90) [49, 50]. On the other hand, Cronbach alpha and McDonald's omega were used to evaluate reliability.

Subsequently, a Bayesian multiple regression analysis [51] was performed to determine the predictive potential of each motivational practice on the motivational orientation towards learning and to estimate the best model from the different variables. This analysis allows us to establish with greater precision the probability of each variable being incorporated into the model. In addition, it provides a credibility interval for each of the point estimators (beta) of the variables retained in the model [52]. The resulting evidence was evaluated based on the criteria used by Kass and Raftery [53], slightly more demanding than those proposed by Harold Jeffreys [54].

The choice for a Bayesian analysis is that although traditional (frequentist) analysis can perform a similar estimation, and the a priori probability is the same for each of the practices considered, the Bayesian option allows dealing with variable selection and uncertainty in a more integrated way [55].

## Results

A Bayesian multiple regression was carried out to determine the extent to which the motivational potential of nine different types of teaching practices (MMPs) (see Table 1) were predictors of motivational orientation towards learning. The motivational potential of the different practices was operationalized as the root of the product of the frequency and the students' perceived motivational effect of each type of practice (see Table 2). In both cases, an uninformed uniform prior [P(M)] = .00195 was set for each possible model. In the analysis, the JZS prior with an r scale of .354 and Bayesian Adaptive Sampling (BAS) was used as the sampling method.

**Table 1. Motivation potential of practices–descriptives.**

| | Health | | Education | |
|---|---|---|---|---|
| | **mean** | **sd** | **Mean** | **sd** |
| Mastery | 4.972 | 0.788 | 5.204 | 0.660 |
| MPP Rewards | 1.871 | 1.538 | 1.706 | 1.644 |
| MPP Punishment | 0.821 | 1.307 | 0.739 | 1.254 |
| MPP Utility | 3.908 | 1.028 | 4.205 | 0.913 |
| MPP Importance | 4.110 | 1.015 | 4.395 | 0.862 |
| MPP Interest | 3.814 | 1.066 | 4.150 | 0.881 |
| MPP Challenge | 3.513 | 1.252 | 4.018 | 0.972 |
| MPP Emotional Support | 3.634 | 1.151 | 4.011 | 0.956 |
| MPP Safety Ambience | 3.730 | 1.100 | 4.134 | 0.926 |
| MPP Autonomy | 3.630 | 1.211 | 4.046 | 1.059 |

Note. MPP = Motivational potential of practices ($\sqrt{Frequency\ of\ practices * Perception\ of\ motivation}$). Health n = 347; Education n = 324

In the health group, there is robust evidence for a regression model that includes motivational practices oriented to utility, importance, and autonomy ($BF_{10}$ = 2.36 x $10^{17}$) compared to the null model. Together, these variables explain 24.1% of the variance of the motivational orientation toward learning (see Table 2).

In the case of the group of students in the area of education, the best model is formed only by the motivational potential of the practices associated with importance and autonomy as predictors of the motivational orientation toward learning (Mastery) ($BF_{10}$ = 7.14 x $10^{7}$). In this case, the regression explains 13.3% of the variance of the motivational orientation towards learning (see Table 2).

In the case of the group of students belonging to health careers, the main predictor of the model corresponds to the motivational potential of practices focused on Importance ß = .203 [CI = .103, .303], SD = .050; with a very low probability of exclusion in the model (P(excl|data) = .004). To this variable, the motivational potential of practices focused on autonomy and utility is added, whose coefficients reach ß = .113 [CI = .046, .179], SD = .033; and ß = .110 [CI = .013, .207], SD = .049; respectively. While these three variables that account for the motivational potential of the practices show a posterior marginal inclusion probability higher than the a priori probability (see Table 3), it is essential to note that in the case of utility, it shows a probability of exclusion that is at the limit of what is acceptable (P(excl|data) = .409).

In the group of education students, the most influential factor is the motivational potential of practices linked to the promotion of importance ß = .179 [CI = .087, .271], SD = .047; and motivational potential of practices associated with the promotion of autonomy ß = .097 [CI = .026, .167], SD = .036. In all cases, the credibility interval is 95%.

As in the case of the Health students, in the Education group, the remaining variables show a marginal posterior inclusion probability higher than the prior probability (see Table 4). In both groups, the variables considered in the model show strong evidence of their effect and low uncertainty about the size of that effect, as evidenced by the credible intervals for the different variables.

**Table 2. Best model–predicting learning motivational orientation (mastery).**

| Group | Models | P(M) | P(M|data) | BF $_M$ | BF $_{10}$ | R$^2$ |
|---|---|---|---|---|---|---|
| Health | *Utility + Importance + Autonomy* | .00195 | .1364 | 80.7 | 2.36e+17 | 0.241 |
| Education | *Importance + Autonomy* | .00195 | .1510 | 90.9 | 7.14e+7 | 0.133 |

**Table 3. Posterior summaries of coefficients (Health).**

| Coefficient | P(incl) | P(excl) | P(incl\|data) | P(excl\|data) | BF $_{incl}$ | Mean | SD | 95% CI | |
|---|---|---|---|---|---|---|---|---|---|
| | | | | | | | | Lower | Upper |
| Intercept | 1.000 | .000 | 1.000 | .000 | 1.00 | 4.972 | 0.0370 | 4.8993 | 5.045 |
| Utility | .500 | .500 | 0.591 | .409 | 1.45 | 0.110 | 0.0492 | 0.0134 | 0.207 |
| Importance | .500 | .500 | 0.996 | .004 | 247.39 | 0.203 | 0.0508 | 0.1031 | 0.303 |
| Autonomy | .500 | .500 | 0.937 | .063 | 14.86 | 0.113 | 0.0338 | 0.0461 | 0.179 |

## Discussion

Searching for practices that have a motivational effect on learning at the university level, we have evaluated the motivational potential of nine teaching practices from the student's perspective. We were interested in a specific aspect: which of them contributed to explaining, not the intensity of motivation for the task, but the specific orientation towards learning.

Only three of the nine practices examined demonstrated a statistically significant influence on the motivational orientation toward learning. Specifically, motivational practices that emphasized the importance, autonomy, and utility of learning were found to have a significant impact. However, the utility variable exhibited significance solely within the health students' group and with a marginal contribution. Consequently, there is a discernible divergence in the effects of these variables between the two groups. Notably, practices centered around reward, punishment, challenge and self-efficacy, emotional support, and creating a safe classroom environment did not exhibit a statistically significant contribution to the model.

How can we explain these two configurations? Why is it that while education students structure their learning orientation mainly in terms of importance and autonomy, health students also incorporate utility-oriented as a significant variable to explain their motivational orientation towards learning?

In the case of future educators, the motivational orientation towards learning (Mastery) is configured thanks to teaching practices that emphasize the importance of what is being learned and activities that favor autonomy, allowing students to develop their mastery of the subject. This type of practice has a positive effect on the student's desire to learn during their professional training. The results show that the motivational orientation towards learning is favored fundamentally by those practices that connect what will be learned with the professional teaching identity. This relationship between importance and identity is consistent with the available evidence [56, 57]. In turn, the results are consistent with evidence pointing to the fact that the practices developed by teachers in the classroom favor autonomy, catalyze high-quality motivations, and foster engagement and, therefore, the adaptive functioning of students in the classroom and their results [58, 59]. Autonomous work would, in turn, allow the internalization of the personal reworking of the professional project and transform it into a source of intrinsic motivation [27].

**Table 4. Posterior summaries of coefficients (Education).**

| Coefficient | P(incl) | P(excl) | P(incl\|data) | P(excl\|data) | BF $_{incl}$ | Mean | SD | 95% C I | |
|---|---|---|---|---|---|---|---|---|---|
| | | | | | | | | Lower | Upper |
| Intercept | 1.000 | .000 | 1.000 | .000 | 1.00 | 5.204 | 0.034 | 5.137 | 5.272 |
| Importance | .500 | .500 | .988 | .012 | 134.23 | 0.179 | 0.047 | 0.087 | 0.271 |
| Autonomy | .500 | .500 | .704 | .296 | 7.00 | 0.103 | 0.038 | 0.028 | 0.178 |

For the case of healthcare careers, utility-oriented practices correspond to the variable with weight in the model. A tentative explanation is that in this professional field, clinical decision-making has immediate repercussions on the patients treated [60], so what is perceived as applicable is professionally structuring. Thus, a motivational practice-oriented to show the utility of learning is indissoluble with professional identity [43]. Likewise, it is essential not to forget that in the case of health, the utility effect is added to the explanatory capacity that the practices of importance and autonomy have, also present in the area of education.

Finally, it is essential to note that the remaining measured variables not retained in the models explaining motivational orientation toward learning may affect motivation in general. However, to favor specifically motivational orientation toward learning, we should privilege motivational teaching practices that insist on the importance of these learnings for their professional identity, these practices that favor autonomy and self-determination, and also these motivational practices focused on showing the utility of such learning.

This study, unlike others, has drawn on more than one theory to identify the factors that influence motivational orientation towards learning. This option is in line with the discussion on using more than one theory to address complex motivational phenomena [24, 25]. This choice of considering constructs from different theories allowed us to identify those practices that impact a motivational orientation focused on learning. In this way, we believe we can contribute, in a practical way, to decision-making on how to stimulate the desire to learn.

In this context, the results show that statistically significant predictors of motivational orientation towards learning come from two different theories: Expectancy-Value and Self-Determination. While Utility and Importance correspond to key elements within the notion of task value, proposed by the Expectancy-Value theory [61], Autonomy is a fundamental variable for understanding motivated behavior within the theoretical proposal of Self-Determination Theory [5, 62].

In this study, we wanted to analyze the effect of teachers' motivational practices on their motivational orientation toward learning. The results show that the percentage of variance does not exceed 25%. Even though these data show that there would be another 75% unexplained, this percentage of explained variance accounts for more directly modifiable variables.

Although other variables influence this type of motivational orientation, they are not necessarily the object of direct intervention in university teaching. Psychological or personality variables, previous experiences, and cultural aspects escape our interest in identifying variables whose modification through teaching can benefit the desire to learn in-depth the disciplinary and professional contents that the educational institution intends. Comparing these effects in health careers (moderate effect, d = .564) and education (small effect, d = .357) with the mean effect size of motivational factors toward learning reported by Hattie [1], we observe that this contribution is close to the mean effect of motivational variables (d = 0.48).

The relevance of these findings lies in identifying specific and differential aspects between these two groups of students concerning the practices that would favor motivational orientation towards learning. These findings are consistent with the advances of some motivational theories toward a situated character in motivational studies [63].

At a practical level, knowing the formative particularities in the various careers, these results also invite us to identify in each area of professional training the teaching practices whose motivational potential favors motivational orientation towards learning.

## Limitations and perspectives

This study addressed the motivational potential of teaching practices based on student reports. The ideal would have been to conduct an observational study of the frequency of teaching practices and match it with each participant. Given the interest in exploring in such a way that

we could contrast this phenomenon in health and education careers, the option of direct observation of teaching practices was clearly unfeasible. The diversity of careers and courses within them made an analysis that could match each of the 730 students with their respective professors unfeasible.

Along these lines, it is necessary to understand more deeply how some teaching practices that potentially favor motivation influence the activation of students' attentional, cognitive, metacognitive, and volitional resources and orient them for learning. The current results contribute to this task, but there are still at least three complementary lines of work ahead: 1) to complement this analysis with qualitative studies, 2) to analyze the mediating and moderating effect of other variables and 3) to advance, based on more complex mathematical models (e.g., Monte Carlo Method simulations) in the modeling of the combined effect of these variables.

## Conclusion

In this study we found strong evidence of the specific effect of motivational practices on the motivational orientation for learning. In health programs, teaching practices related to utility, importance, and autonomy constitute predictor variables of the motivational orientation toward learning. In contrast, teaching practices that predict learning motivation orientation in education programs are importance and autonomy.

## Supporting information

**S1 Appendix. Items from the inventory to measure motivational potential of motivational teaching practices (Original version in Spanish).**
(DOCX)

## Author Contributions

**Conceptualization:** Jorge Valenzuela, Jorge Miranda-Ossandon, Carla Muñoz, Andrea Precht, Milenko Del Valle.

**Data curation:** Jorge Valenzuela, Carla Muñoz.

**Formal analysis:** Jorge Valenzuela, Jorge Miranda-Ossandon, Carla Muñoz, Andrea Precht, Milenko Del Valle, Juan-Gabriel Vergaño-Salazar.

**Funding acquisition:** Jorge Valenzuela, Jorge Miranda-Ossandon.

**Investigation:** Jorge Valenzuela, Jorge Miranda-Ossandon, Andrea Precht, Juan-Gabriel Vergaño-Salazar.

**Methodology:** Jorge Valenzuela, Milenko Del Valle.

**Project administration:** Jorge Valenzuela.

**Writing – original draft:** Jorge Valenzuela, Jorge Miranda-Ossandon, Carla Muñoz, Andrea Precht, Milenko Del Valle, Juan-Gabriel Vergaño-Salazar.

**Writing – review & editing:** Jorge Valenzuela, Jorge Miranda-Ossandon, Carla Muñoz, Andrea Precht, Milenko Del Valle, Juan-Gabriel Vergaño-Salazar.

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
