## [Decision Letter · Decision Letter 0]

16 Nov 2023

PONE-D-23-30806Learning-oriented motivation: Examining the impact of teaching practices with motivational potentialPLOS ONE

Dear Dr. Valenzuela,

Thank you for submitting your manuscript to PLOS ONE. After careful consideration, we feel that it has merit but does not fully meet PLOS ONE’s publication criteria as it currently stands. Therefore, we invite you to submit a revised version of the manuscript that addresses the points raised during the review process.

We look forward to receiving your revised manuscript.

Kind regards,

Henri Tilga, PhD

Academic Editor

PLOS ONE

Journal Requirements:

Additional Editor Comments:

The Reviewers have provided several useful comments to increase the quality of this manuscript. Please carefully follow all the comments made by the Reviewers and revise the manuscript accordingly.

Reviewers' comments:

Reviewer's Responses to Questions

**Comments to the Author**

1. Is the manuscript technically sound, and do the data support the conclusions?

Reviewer #1: Yes

Reviewer #2: Partly

2. Has the statistical analysis been performed appropriately and rigorously? 

Reviewer #1: Yes

Reviewer #2: Yes

3. Have the authors made all data underlying the findings in their manuscript fully available?

Reviewer #1: No

Reviewer #2: Yes

4. Is the manuscript presented in an intelligible fashion and written in standard English?

Reviewer #1: Yes

Reviewer #2: Yes

5. Review Comments to the Author

Reviewer #1: In this work an interesting study is presented to determine which motivational practices are the most effective in guiding students towards focusing on learning, not only obtaining good grades. I am grateful for the opportunity to review this manuscript. I find the research question interesting and valuable because there are many different strategies in many theoretical motivational frameworks to motivate students. However, in higher education the motivation to graduate should not be the only important aspect to consider. Educators and students should aim towards deeper learning, the real integration of skills and knowledge gained during the studies.

The authors have extracted motivational practices from different motivational theories and compared the effect they have on the motivational orientation of health and education students. It was revealed that depending on the study program, the effective strategies to use may vary. The most important predictors of motivation towards learning were fostering autonomy and understanding the importance of studies for both programs. In the case of health students, usefulness of the studies was an additional component of the predictive model. The other motivational practices assessed may contribute to supporting motivation in general.

The manuscript has several strengths. The research question is original, the integration of different theoretical frameworks enables finding the most effective practices, and the large number of participants adds reliability. There are also some aspects of the manuscript that can be improved.

1. In the first paragraph of introduction, the authors bring in the term motivational orientation. Please provide a definition of motivational orientation.

2. Regarding the introduction, it would be beneficial to expand on the theoretical frameworks that form the basis of the study. Consider providing a brief explanation of each and their contribution to understanding the motivation to learn.

3. At the end of introduction, the authors mention that current literature shows the benefits of integrating different theories. Please explain in more detail what those benefits are.

4. The same aspect can be covered in more detail in discussion (rows 259-261). As integrating theories is one of the strengths of the study, it would be beneficial to provide more information about the specific findings of the cited studies and their relevance to the current research.

5. In the methods section, I find it necessary to clarify how the questionnaire was developed. Were the items used to measure the motivational potential of practices based on previously used instruments? I would encourage authors to make the administered questionnaire available. Also, please include information about the verification of the psychometric characteristics of the scales (e.g., in an additional file).

As to the readability of the manuscript, here are some recommendations for minor revisions:

a. It would be helpful to use the same terminology throughout the text (e.g., utility vs usefulness). Do learning orientation (row 18) and motivational orientation have the same meaning?

b. Some of the sentences could be better worded for clear understanding (e.g., rows 22-24 “While the effects…”; rows 95-98 “In this context…”).

c. In discussion, the paragraph in rows 223-230 lists the three significant practices twice. Please consider removing the repetition.

d. All in all, the language of the manuscript is good. However, there are some sentences that could be improved to make the text easier to follow. I would recommend asking a native English speaker to check the wording of the manuscript.

Reviewer #2: It is a relevant research, focused on evidencing motivational practices in the motivational orientation for learning. The literature review is appropriate and well-founded to understand the research. The statistical methods are robust and well-grounded. The results are suitable for what is pursued in the research, however, they lack robustness.

It would be recommended to clarify certain issues:

1. The sample consists of Chilean university students. At this point, it is recommended to declare the size of the population, differentiated by gender, the different types of universities that exist in Chile. Characterize the population of university students.

2. Characterize the sample, indicating the percentage it represents of the population, for each type of university, regions, etc. This is relevant to have clarity on who your sample is representing.

3. The value of the Determination Coefficient is striking, as it does not exceed 25%. It should be explained that there is more than 75% of the variability of the dependent variable that is not being explained by the variables considered in the study and conduct a discussion on the potential of these findings, which are rather limited.

4. In Table 1, it is suggested to remove the column of sample sizes for health and education and refer to them in the text.

6. PLOS authors have the option to publish the peer review history of their article (what does this mean?). If published, this will include your full peer review and any attached files.

Reviewer #1: No

Reviewer #2: No

---

## [Author Response · Author response to Decision Letter 0]

1 Jan 2024

Response to reviewers

We want to thank the reviewers for their comments and suggestions for improving the manuscript. We attach the details of the changes made to consider all the reviewers' suggestions.

Some other changes were made, such as including references or changes in writing.

The link to access the database is also added, and the original questionnaire used to assess the motivational potential of teaching practices is annexed.

Editor

1. Please ensure that your manuscript meets PLOS ONE's style requirements, including those for file naming. The PLOS ONE style templates 

We have revised the manuscript and corrected from the PLOSONE standards.

The funding sources have been corrected by specifying the grant number and the researcher receiving the grant.

3. In your Data Availability statement, you have not specified where the minimal data set underlying the results described in your manuscript can be found.

The data underlying the results of this article are available at: https://doi.org/10.5281/zenodo.10427851

Reviewer 1

1. In the first paragraph of introduction, the authors bring in the term motivational orientation. Please provide a definition of motivational orientation.

A definition of motivational orientation is incorporated and made explicit in the first paragraph of the introduction.

2. Regarding the introduction, it would be beneficial to expand on the theoretical frameworks that form the basis of the study. Consider providing a brief explanation of each and their contribution to understanding the motivation to learn. 

The contribution that theories make to the understanding of motivation to learn is made explicit in the introduction.

3. At the end of introduction, the authors mention that current literature shows the benefits of integrating different theories. Please explain in more detail what those benefits are. 

It makes explicit the meaning and benefits of integrating different theories. (See rows 92 et seq.)

4. The same aspect can be covered in more detail in discussion (rows 259-261). As integrating theories is one of the strengths of the study, it would be beneficial to provide more information about the specific findings of the cited studies and their relevance to the current research. 

This aspect is covered in discussion (see rows 298 et seq.)

5. In the methods section, I find it necessary to clarify how the questionnaire was developed. Were the items used to measure the motivational potential of practices based on previously used instruments? I would encourage authors to make the administered questionnaire available. Also, please include information about the verification of the psychometric characteristics of the scales (e.g., in an additional file).

The origin of the instrument is explained, and the questionnaire is annexed. See appendix S1

a. It would be helpful to use the same terminology throughout the text (e.g., utility vs usefulness). Do learning orientation (row 18) and motivational orientation have the same meaning?

To avoid confusion, the terminology "utility" is homogenized throughout the text.

In the case of "learning orientation" and "motivational orientation, the expression is homogenized as "orientation towards learning" throughout the text, and it is made explicit that this is a specific case of motivational orientation).

b. Some of the sentences could be better worded for clear understanding (e.g., rows 22-24 “While the effects…”; rows 95-98 “In this context…”).

B. About rows 22-24 the paragraph is rewrite:

(see rows 27 et seq.)

Although the impact on motivational orientation toward learning may vary slightly across different fields of study, the primary predictors consistently are practices that emphasize importance and foster autonomy. The effect of utility-focused motivational practices is observed only within the predictive model for the group of health students.

C. In discussion, the paragraph in rows 223-230 lists the three significant practices twice. Please consider removing the repetition.

This text was reformulated: (see rows 260 et seq.):

Only three of the nine practices examined demonstrated a statistically significant influence on the motivational orientation toward learning. Specifically, motivational practices that emphasized the importance, autonomy, and utility of learning were found to have a significant impact. However, the utility variable exhibited significance solely within the health students' group and with a marginal contribution. Consequently, there is a discernible divergence in the effects of these variables between the two groups. Notably, practices centered around reward, punishment, challenge and self-efficacy, emotional support, and creating a safe classroom environment did not exhibit a statistically significant contribution to the model.

D. All in all, the language of the manuscript is good. However, there are some sentences that could be improved to make the text easier to follow. I would recommend asking a native English speaker to check the wording of the manuscript. 

The suggestion was welcome. 

Reviewer 2

1.The sample consists of Chilean university students. At this point, it is recommended to declare the size of the population, differentiated by gender, the different types of universities that exist in Chile. Characterize the population of university students.

A summary of the university education system in Chile is added in the “participants” section.

The fact that the Chilean educational system includes both public and private universities is made explicit, and the configuration of the participants is detailed. 

(See Participants section)

2. Characterize the sample, indicating the percentage it represents of the population, for each type of university, regions, etc. This is relevant to have clarity on who your sample is representing.

A more precise characterization of the participants was carried out. 

3. The value of the Determination Coefficient is striking, as it does not exceed 25%. It should be explained that there is more than 75% of the variability of the dependent variable that is not being explained by the variables considered in the study and conduct a discussion on the potential of these findings, which are rather limited.

A commentary focused on this aspect (the effect size observed in this study) is included in the discussion section, comparing it with the results of Hattie's meta-analysis. (See rows 307 et seq).

4. In Table 1, it is suggested to remove the column of sample sizes for health and education and refer to them in the text.

The column of sample sizes for health and education were removed. The number of participants is reported in footnote.

---

## [Decision Letter · Decision Letter 1]

15 Jan 2024

Learning-oriented motivation: Examining the impact of teaching practices with motivational potential

PONE-D-23-30806R1

Dear Dr. Valenzuela,

We’re pleased to inform you that your manuscript has been judged scientifically suitable for publication and will be formally accepted for publication once it meets all outstanding technical requirements.

Kind regards,

Henri Tilga, PhD

Academic Editor

PLOS ONE

Additional Editor Comments (optional):

Reviewers' comments:

Reviewer's Responses to Questions

**Comments to the Author**

1. If the authors have adequately addressed your comments raised in a previous round of review and you feel that this manuscript is now acceptable for publication, you may indicate that here to bypass the “Comments to the Author” section, enter your conflict of interest statement in the “Confidential to Editor” section, and submit your "Accept" recommendation.

Reviewer #1: (No Response)

2. Is the manuscript technically sound, and do the data support the conclusions?

Reviewer #1: Yes

3. Has the statistical analysis been performed appropriately and rigorously? 

Reviewer #1: Yes

4. Have the authors made all data underlying the findings in their manuscript fully available?

Reviewer #1: Yes

5. Is the manuscript presented in an intelligible fashion and written in standard English?

Reviewer #1: Yes

6. Review Comments to the Author

Reviewer #1: (No Response)

7. PLOS authors have the option to publish the peer review history of their article (what does this mean?). If published, this will include your full peer review and any attached files.

Reviewer #1: No

---

## [Editor Report · Acceptance letter]

16 Feb 2024

PONE-D-23-30806R1 

PLOS ONE

Dear Dr. Valenzuela, 

I'm pleased to inform you that your manuscript has been deemed suitable for publication in PLOS ONE. Congratulations! Your manuscript is now being handed over to our production team.

Kind regards, 

on behalf of

Dr. Henri Tilga 

Academic Editor

PLOS ONE